# Molecular architecture of softwood revealed by solid-state NMR

Oliver M. Terrett [1], Jan J. Lyczakowski [1,2], Li Yu [1,2], Dinu Iuga[3], W. Trent Franks [3], Steven P. Brown [3], Ray Dupree[3]* & Paul Dupree [1,2]*

Economically important softwood from conifers is mainly composed of the polysaccharides cellulose, galactoglucomannan and xylan, and the phenolic polymer, lignin. The interactions between these polymers lead to wood mechanical strength and must be overcome in biorefining. Here, we use $^{13}C$ multidimensional solid-state NMR to analyse the polymer interactions in never-dried cell walls of the softwood, spruce. In contrast to some earlier softwood cell wall models, most of the xylan binds to cellulose in the two-fold screw conformation. Moreover, galactoglucomannan alters its conformation by intimately binding to the surface of cellulose microfibrils in a semi-crystalline fashion. Some galactoglucomannan and xylan bind to the same cellulose microfibrils, and lignin is associated with both of these cellulose-bound polysaccharides. We propose a model of softwood molecular architecture which explains the origin of the different cellulose environments observed in the NMR experiments. Our model will assist strategies for improving wood usage in a sustainable bioeconomy.

---

[1] Department of Biochemistry, University of Cambridge, Hopkins Building, The Downing Site, Tennis Court Road, Cambridge CB2 1QW, UK. [2] Natural Material Innovation Centre, University of Cambridge, 1 Scroope Terrace, Cambridge CB2 1PX, UK. [3] Department of Physics, University of Warwick, Coventry CV4 7AL, UK. *email: rayd.warwick@gmail.com; pd101@cam.ac.uk

The majority of biomass carbon is stored by plants in forests as wood[1,2]. Conifers make up the majority of trees in boreal forests, which form 30% of forest area, and the majority of commercially planted trees[3,4]. Softwood is therefore economically and ecologically important, especially in the Northern hemisphere. Conifers form the largest natural renewable resource for producing fuels, construction materials and chemical feedstocks, but softwood properties are not optimal for each application. To improve processing of timber, and to breed trees with tailored properties, it is important to understand how the molecular constituents of wood come together to form this versatile and recalcitrant material.

Softwood is predominantly composed of water-conducting tracheids which are surrounded by secondary cell walls. These secondary walls, at the molecular level, are made primarily out of the polysaccharides cellulose, xylan and galactoglucomannan (GGM) and the polyphenolic polymer lignin[5]. We do not yet fully understand how these constituents come together to form the cell wall matrix. At the microscopic level, these cell wall components form structures known as macrofibrils[6]. FT-IR experiments indicate that the hemicelluloses, and to some extent the lignin, are aligned in parallel to the cellulose, perhaps within these softwood macrofibrils[7,8]. Our recent analysis using cryoSEM, along with previous papers using similar techniques[9,10], indicates that in softwood cell walls the macrofibrils can be up to 60 nm in diameter and that in secondary cell walls of a model hardwood species, Arabidopsis, smaller macrofibrils,are composed of cellulose, xylan and lignin[11]. These experiments provide insight into the mesoscale organisation of the cell wall, however, they cannot yet provide sub-nanometre resolution information about the direct molecular interactions between cell wall components in softwoods. In addition, these techniques cannot definitively discriminate between cell wall components. Solid-state NMR provides precise conformational, chemical and proximity information and so can address this gap in our understanding of cell wall assembly by providing sub-nanometre constraints on the interactions between cell wall polymers[12–14].

The molecular structure of cell wall components strongly influences their interactions. Cellulose elementary fibrils are composed of β-1,4 glucan chains that each form a flat two-fold screw ribbon, where each glucosyl residue is rotated 180° relative to the previous residue[15]. The crystalline arrangement of these glucan chains generates a microfibril, which in spruce is predicted to have a diameter of between 3 and 4 nm[16], with surfaces that diverge in hydrophilicity. Consequently, the other wall components GGM, xylan and lignin may interact with the cellulose microfibril faces differently.

Solid-state NMR has been used to identify and interpret the substantial heterogeneity of glucosyl residue environments in cellulose[17]. Two sets of environments have been identified in plant cellulose, sometimes called domain 1 and domain 2, with distinctive chemical shift differences at carbons 4 and 6[18,19]. Domain 1 glucosyl residue environments have carbon 4 chemical shifts around 89 ppm and carbon 6 chemical shifts around 65 ppm. Domain 2 environments have carbon 4 shifts around 84 ppm and carbon 6 chemical shifts of 62 ppm. These domains are often assigned to residues in interior and surface chains of the microfibrils. The assignment of domain 1 to interior chains seems inconsistent with their reported proximity with the hemicellulose xylan and pectic polysaccharides in Arabidopsis[12,14]. Structural factors, including the conformation of the carbon 6 hydroxymethyl group, are thought to cause the chemical shift differences between domains 1 and 2[20,21]. The altered hydroxymethyl conformation in domains 1 and 2 may reflect different interactions between cellulose chains and also between cellulose and other cell wall components.

Xylan forms 5-15% of conifer secondary cell walls[5]. The backbone is composed of β-1,4 linked xylosyl residues with α-1,2 glucuronosyl and α-1,3 arabinosyl substitutions[22]. Recently solid-state NMR has shown that xylan binds to cellulose and adopts a two-fold screw conformation in the eudicot Arabidopsis[14]. Furthermore, in Arabidopsis, this two-fold screw formation requires an even pattern of xylan substitutions, providing the first evidence that xylan binds to the hydrophilic faces of the fibril[23]. In conifers, most glucuronic acid substitutions are six xylosyl residues apart, with an arabinose substitution two xylosyl residues towards the non-reducing end from the glucuronic acid[24]. The evenly spaced substitutions mean that conifer xylan in a two-fold screw conformation would have an effectively unsubstituted face, and this could enable binding to the hydrophilic face of cellulose[25]. However, multiple studies suggest that xylan does not bind directly to cellulose in conifers, and instead associates with lignin away from cellulose[7,9].

In conifers, GGM forms 15–25% of secondary cell walls[5]. The GGM backbone is β-1,4 linked, and is composed of 75-80% mannosyl residues with 20–25% glucosyl residues[26]. Between 30 and 50% of the mannosyl residues are acetylated at carbons 2 or 3 and GGM is thought to be the only acetylated hemicellulose in conifers[22]. The mannosyl residues can also be α-1,6 galactosylated[27]. GGM substitutions are not patterned, as is the case for xylan[26]. Studies of intact walls using microscopy and FT-IR suggest that GGM is bound to cellulose[7,9,28,29]. On the other hand, simulations indicate that the axial position of the carbon 2 hydroxyl and substitution of hydroxyls may prevent binding to certain cellulose faces[30]. Indeed, in vitro studies suggest more highly substituted or acetylated GGMs bind more poorly to cellulose than less branched or de-branched GGMs[28,29]. In a similar fashion to cellulose, de-acetylated konjac glucomannan and unbranched mannans such as those from ivory nut can form crystalline structures with each mannan chain forming a flat ribbon[31–33]. The in muro molecular interaction of GGM with cellulose remains an open question. Here, in muro refers to the native location of wood polysaccharides and lignin in the plant cell wall, in contrast to extracted in vitro polysaccharides.

Gymnosperm lignin is composed mainly of the monolignol coniferyl alcohol, which has just one methoxyl on the phenyl ring[34]. Monolignols are converted to radicals by laccases/peroxidases in muro and polymerise via combinatorial coupling. Covalent linkages between lignin and polysaccharides form through several mechanisms, but the spatial relationship between lignins and polysaccharides has not been investigated until recently[35,36]. Some studies suggest that lignin is aligned along the axis of cellulose microfibrils, and that xylan associates with lignin separately from galactoglucomannan and cellulose in gymnosperm cell walls[7,9]. Recently, in grass and Arabidopsis cell walls, solid-state NMR demonstrated the existence of extensive noncovalent interactions between lignin and xylan[37]. This included the three-fold and a modified two-fold conformations of xylan[37]. Interestingly, the lignin composition may affect interactions with polysaccharides, as it was suggested that more highly methylated syringyl units associate more strongly with xylan in grasses[37]. It is not clear if this is due to the temporal patterning of lignification, or a stronger biophysical interaction. Establishing whether lignin interacts with both xylan and glucomannan in spruce and whether the unique composition of gymnosperm lignin affects any interaction is an important question.

In this study, we use multidimensional $^{13}C$ solid-state magic angle spinning (MAS) NMR to investigate the in muro interactions of GGM and xylan with cellulose and lignin in never-dried wood of a conifer species, Picea abies (Norway spruce). We find that GGM interacts with the cellulose microfibril surface. The similarity of the chemical shifts of GGM and those

semi-crystalline mannans suggests the GGM may form a flattened ribbon on the microfibril surface. We find that, as in Arabidopsis, xylan forms a two-fold screw bound to cellulose, revealing that this mode of xylan-binding to cellulose is widespread in vascular plants. GGM and xylan are also close to each other when bound to cellulose; we propose that xylan and GGM both coat the same cellulose microfibrils. Finally, we show that some lignin is closely associated with the polysaccharides. These are important advances in our understanding of the structure of secondary cell walls in conifers.

## Results

**Assignment of polysaccharides in spruce secondary cell walls.** To gain an overview of the rigid polysaccharides in spruce secondary cell walls, we analysed [13]C-enriched spruce wood by solid-state NMR. The CP-INADEQUATE experiment shows correlated peaks between two covalently bonded carbons, each of which have a single quantum (SQ) and double quantum (DQ) chemical shift. The DQ shift, which is the sum of the SQ shifts of the two bonded carbons, resolves overlapping pairs of signals in a second dimension. The carbohydrate region of the spectrum, shown in Fig. 1, is dominated by cellulose signals, while the pectin and xyloglucan signals are absent, reflecting the monosaccharide composition of this sample, Supplementary Table 1[5]. The presence of abundant mannose and xylose in the monosaccharide analysis further indicated the spruce wood sample had a high proportion of secondary cell wall polysaccharides, specifically GGM and xylan.

Previously, we assigned cellulose environments in Arabidopsis with a carbon 4 (C4) shift of ~89 ppm as cellulose domain 1, and those with a C4 shift of ~84 ppm as cellulose domain 2[18]. At least six glucose environments in cellulose are partially resolved in this spruce spectrum, and they are labelled as domains 1A–C and 2A–C. Multiple glucose environments have also been reported in ssNMR spectra of cellulose in other plant cell walls[19]. Xylan peaks are clearly visible in the spectrum. Interestingly, both two-fold and three-fold xylan carbon 4 (Xn4) and carbon 5 (Xn5) peaks are present. Although previous models suggested xylan is not bound to cellulose in gymnosperms[7,9], the presence of two-fold xylan suggests that this xylan is bound to cellulose, as recently demonstrated in Arabidopsis[14,23].

Two pairs of carbon 1 and carbon 2 peaks, both at ~100.9/ 101.9 and 71.9/72.0 ppm, respectively, are present in the spectrum. Due to the high amount of GGM in spruce (Supplementary Table 1)[5], and their similarity to solution NMR shifts of GGM (Supplementary Table 2), we hypothesised that these peaks arise from mannosyl residues in this polysaccharide. In addition, these shifts are consistent with those of semi-crystalline ivory nut mannan and cellulose-bound GGM from in vitro bacterial cellulose composites[28,31]. The carbon 1 at 100.9 ppm was also previously assigned to hemicellulose in a one-dimensional solid-state MAS NMR spectrum of Sitka spruce[38]. To support this assignment, we investigated CP-INADEQUATE spectra of stems of wild type and mutant Arabidopsis plants. The Arabidopsis *csla2/3/9* triple mutant in Cellulose Synthase-Like A (CSLA) genes has no GGM in its secondary cell walls, but grows normally[39]. The proposed pair of coupled GGM mannosyl carbon 1 (M1) and M2 peaks were present in the spectrum of wild type Arabidopsis, which contain 1-2% GGM, but absent from the spectrum of the *csla2/3/9* GGM-deficient mutant (Fig. 2a) confirming the two distinct M1 and M2 assignments.

To assign further GGM mannosyl residue shifts in muro, we performed a short mixing time (30 ms) CP-PDSD experiment on spruce. PDSD experiments utilise through-space transfer of magnetisation (dipolar coupling) between [13]C nuclei. At this

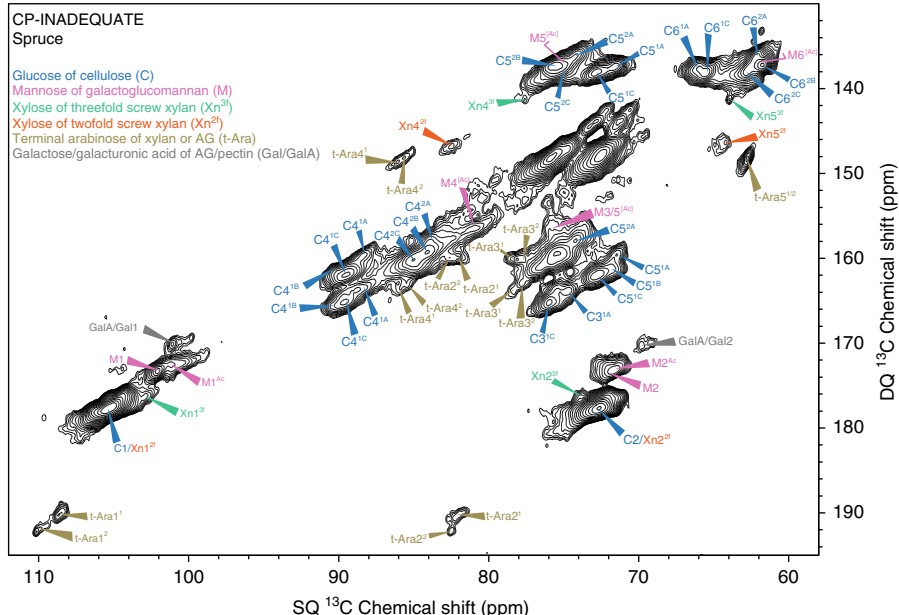

**Fig. 1** An overview of immobile polysaccharides in spruce cell walls. The carbohydrate region of a refocussed CP-INADEQUATE [13]C MAS NMR spectrum of [13]C enriched spruce wood. Most carbons in the major polysaccharides cellulose, galactoglucomannan (GGM) and xylan are labelled, as are the terminal arabinose of xylan or arabinogalactans (AG) and galactose/galacturonic acid residues of arabinogalactans or pectin. The two terminal arabinose residues are labelled 1 and 2 to differentiate them. For cellulose, the environments have been split into two groups, domain 1 and 2, as in[18]. Here, each cellulose domain is further resolved into environments. For GGM, the acetylated mannosyl residue carbons are labelled Ac, where they share the same chemical shift as the unacetylated mannose carbons, they are labelled [Ac]. The positions of GGM carbons 5–6 are obscured by cellulose peaks in the same region in the INADEQUATE spectrum but can be determined from a 30 ms PDSD experiment. Their positions, see full assignments in Supplementary Table 2, are labelled with an unfilled arrow. The region with single quantum (SQ) = 70–80 ppm and double quantum (DQ) = 142–152 ppm is unlabelled due to overlapping peaks from multiple polysaccharides

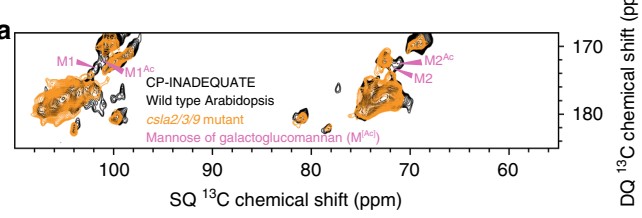

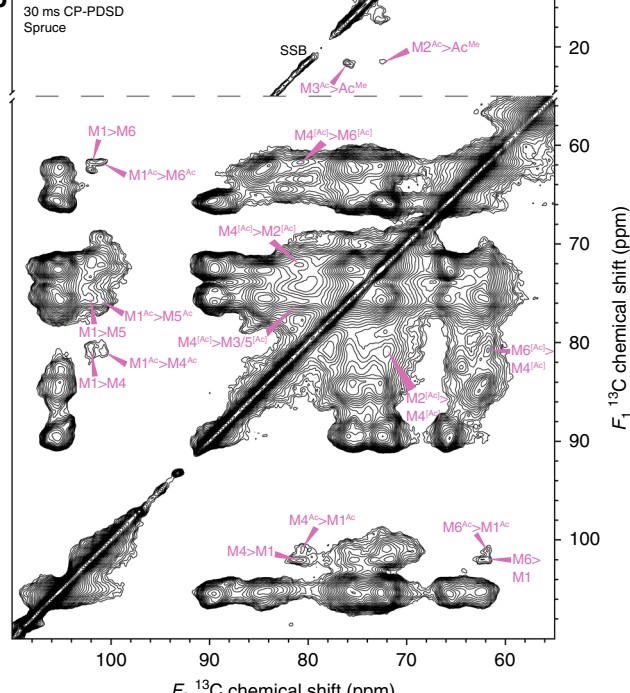

**Fig. 2** Assigning the chemical shifts of GGM in spruce. **a** The carbon 1-2 region of $^{13}C$ CP-INADEQUATE MAS NMR spectra of wild type Arabidopsis and the *csla2/3/9* mutant. The GGM carbons 1 and 2 are labelled. The wild type spectrum was previously published in (Grantham et al., 2017)[23]. **b** The acetate methyl (Ac$^{Me}$) and carbohydrate regions of a 30 ms mixing time $^{13}C$ CP-PDSD MAS NMR spectra of spruce is shown. The intramolecular cross-peaks of two GGM mannose backbone residues are labelled. For GGM, the acetylated mannosyl residue carbons are labelled Ac; where they share the same chemical shift as the unacetylated mannose carbons they are labelled [Ac]. Spinning side bands are marked SSB

**Table 1 In muro chemical shifts of spruce galactoglucomannan**

|         | 1     | 2    | 3    | 4    | 5    | 6    |
|---------|-------|------|------|------|------|------|
| M       | 101.9 | 72.0 | –    | 80.4 | 75.8 | 61.6 |
| M$^{Ac}$ | 100.9 | 71.9 | 75.9 | 80.4 | 75.8 | 61.6 |

The $^{13}C$ chemical shifts in ppm of the two GGM mannosyl residues are summarised. M denotes the unacetylated mannosyl residue, while the superscript Ac denotes the acetylated mannosyl residue

solution NMR of GGM backbone mannosyl residues (Supplementary Table 2). However, both the M4 and M4$^{Ac}$ (M4$^{[Ac]}$) shifts of 80.4 ppm are 2-13 ppm higher than the solution NMR shifts of 67.6-78.5 ppm. Such a change in M4 shift suggests that GGM adopts an altered in muro conformation, possibly when bound to cellulose, as seen when xylan is bound to cellulose[14]. Interestingly, the carbon 1, 4 and 6 shifts are similar to those of the mannan I crystal allomorph (Supplementary Table 2), which can be formed by unbranched mannans[31,32]. In this crystal allomorph, the mannosyl residues are in a two-fold screw conformation, similar to the glucan chains of cellulose, suggesting that the branched GGM in spruce cell walls may form a two-fold screw. The in muro spruce GGM chemical shifts do differ significantly from mannan I at carbons 2 and 3, but this could be due to the lack of acetyl substitution on carbons 2 and 3 in the mannan I crystal.

**Xylan and GGM bind to the same cellulose microfibrils**. To investigate whether the xylan and GGM are bound to the cellulose microfibril surface in conifers, we looked for carbon-carbon cross peaks indicating proximity between these wall components, using further through-space PDSD experiments. We performed CP-PDSD experiments with 100, 400, 1000 and 1500 ms mixing times. As the mixing time increases, the magnetisation can transfer through space between nuclei that are further apart up to a maximal distance of ~5–10 Å, or about 25% of the width of a spruce cellulose microfibril[16].

Firstly, we sought to establish whether cross-peaks between two-fold xylan and cellulose domains 1 or 2 could be seen. In the 100 ms mixing time CP-PDSD spectrum, Supplementary Fig. 4, there are cross-peaks between several cellulose sub-domains, such as C6$^{2A/B}$ and C4$^{1C}$, and vice versa. Although the magnetisation had travelled far enough to show the proximity of adjacent, highly abundant glucan chains, there are no cross-peaks seen between the lower abundance xylan and cellulose. However, in the 400 ms mixing time spectrum (Fig. 3) there are intermolecular cross-peaks between Xn4$^{2f}$ (82.4 ppm) and cellulose C4$^{1C}$ (89.5 ppm), C6$^{1C}$ (65.5 ppm) and a weaker cross-peak between Xn4$^{2f}$ and C6$^{2C}$ (62.6 ppm). This shows that in spruce cell walls two-fold xylan is a similar distance from cellulose as adjacent glucan chains in the microfibril are from each other.

Next, we investigated the GGM interaction with cellulose. As in the case of xylan-cellulose cross-peaks, there are no GGM cross-peaks with cellulose in the 100 ms mixing time spectrum. At 400 ms, cross-peaks between M1 (101.9 ppm) and cellulose C1$^{1/2}$ (105 ppm), and C6$^{1C}$ (65.5 ppm) appear (Fig. 3). Since the appearance of hemicellulose to cellulose cross-peaks was also seen for two-fold xylan at this mixing time, this suggests that unacetylated GGM residues are a similar distance from the microfibril surface as two-fold xylan. Interestingly, cross peaks between M1 and both C4$^{1C}$ and C4$^{2A/B}$ (89.5, 83.9 ppm) are seen, indicating both domains of cellulose are similarly close to GGM molecules. To determine whether acetylated GGM residues are close to cellulose as well as the unacetylated GGM, we inspected

short mixing time, cross-peaks occur between spatially close carbons, such as those within the same sugar residue. We used the two M1 chemical shifts to identify cross-peaks between M1 and M4 (80.4 ppm), M5 (75.8 ppm) and M6 (61.6 ppm), Fig. 2b, of the two GGM residues.

Since GGM is the major acetylated cell wall hemicellulose in conifers, carrying acetyl groups (Ac) on M2 and/or M3 on up to 50% of mannosyl residues[26], we hypothesised that the difference between the two mannosyl environments may be acetylation at M2 or M3. We used proximity to the acetate methyl carbon (Ac$^{Me}$) at 21.4 ppm to assign the GGM acetylated mannosyl residue M2$^{Ac}$ (71.9 ppm) and M3$^{Ac}$ (75.9 ppm), Fig. 2b. There is no cross-peak between Ac$^{Me}$ and either M1 at this mixing time. However, in the 100 ms mixing time spectra, a cross-peak between Ac$^{Me}$ and the M1 at 100.9 ppm is present (Supplementary Fig. 3), and so this was assigned as M1$^{Ac}$. The assigned chemical shifts of both the unacetylated mannosyl residue and acetylated mannosyl residue are shown in Table 1. The carbon 2, 3, 5 and 6 shifts are generally similar to those reported for

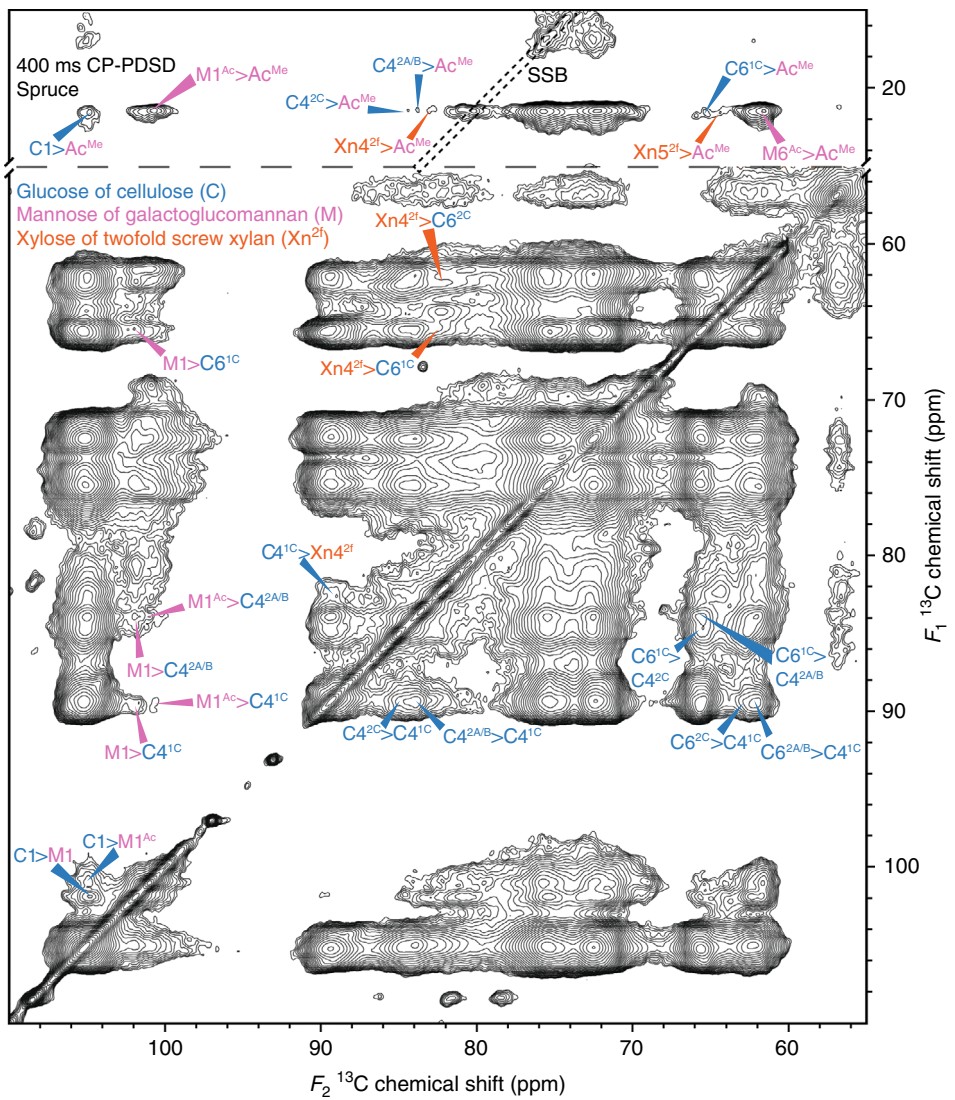

**Fig. 3** Xylan and GGM are a similar distance from the cellulose microfibril surface as glucan chains are from each other. The acetate methyl and carbohydrate regions of a 400 ms mixing time CP-PDSD $^{13}$C MAS NMR spectrum of spruce is shown. Cross-peaks between the different cellulose domains are labelled on the right hand side of the diagonal line in the carbohydrate region. Cross-peaks between cellulose and xylan/GGM are labelled on the left hand side of the diagonal line in the carbohydrate region and in the acetate methyl region of the spectra. Spinning side bands are surrounded by black dotted lines and are marked SSB

the spectra for cross-peaks between the polysaccharide carbons and the acetate methyl group at 21.4 ppm (Ac$^{Me}$). Figure 3 also shows the Ac$^{Me}$ region of the 400 ms mixing time spectrum. In this region, cross-peaks between C1$^{1/2}$, C4$^{2A/B/C}$, C6$^{1C}$ and the GGM acetate methyl group are present. Combined with the similarly long T$_1$s of cellulose (Supplementary Table 3) and GGM carbons, this demonstrates that a substantial component of both acetylated and unacetylated GGM residues are bound to the cellulose surface. Nevertheless, there are subtle differences between the cellulose environments that unacetylated and acetylated mannosyl residues are associated with. For M1, there are cross-peaks to C4$^{1C}$, C4$^{2A/B}$ and C6$^{1C}$ but for Ac$^{Me}$ there are cross peaks to C4$^{2A/B/C}$ and not C4$^{1C}$, though the cross-peak to C6$^{1C}$ suggests that there is proximity to this cellulose environment.

There are cross-peaks between Xn4$^{2f}$ and Xn5$^{2f}$ to Ac$^{Me}$ in the 400 ms mixing time spectra, suggesting that two-fold xylan and GGM are not fully segregated in the cell wall, as has been previously suggested[7] (Fig. 3). To test further the idea that xylan

and GGM are interacting with the same cellulose microfibrils, longer mixing time CP-PDSD spectra were obtained. Figure 4 shows regions of a 1500 ms CP-PDSD spectrum where clear cross peaks from Xn4$^{2f}$ to Ac$^{Me}$ and M1 are seen. Since a longer mixing time is required for the xylan:GGM cross peaks to appear than for cellulose:hemicellulose cross peaks, this suggests that GGM and xylan are closer to the cellulose microfibril than they are to each other. Nevertheless, since the cross peaks are detected, some of these hemicelluloses are within 5–10 Å of each other and therefore probably bound to the same microfibrils. The cross-peaks between xylan and GGM and between these hemicelluloses and cellulose are of similar intensity. This suggests that a large proportion of cellulose-bound xylan and GGM are bound to the same microfibrils.

**Some lignin is close to hemicelluloses and cellulose**. To determine whether lignin is close to cell wall polysaccharides, we examined the CP-PDSD spectra for cross-peaks between polysaccharide carbons and the lignin methoxyl at 56.5 ppm

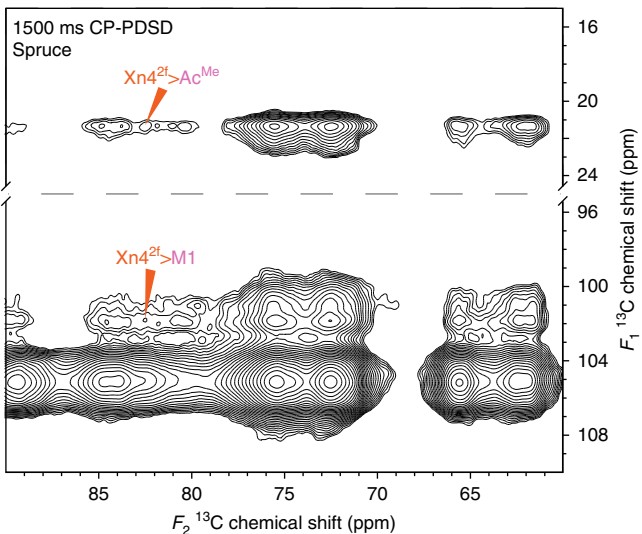

**Fig. 4** Xylan and GGM bind to the same cellulose microfibrils. Acetate methyl and carbon 1 region of a 1500 ms mixing time $^{13}$C CP-PDSD MAS NMR spectrum of spruce. Cross-peaks between xylan and GGM are labelled

(lignin CH$_3$O–), and vice versa. Figure 5a shows the 1500 ms CP-PDSD spectrum together with the 56.5 ppm slices of spectra at various mixing times, Fig. 5b. The 56.5 ppm slices from different mixing times are all normalised to the height of the 56.5 ppm peak. Until 1000 ms, the only cross-peaks are probably intramolecular, from the lignin CH$_3$O- to aliphatic lignin carbons in the range 62–90 ppm that are chemically similar to polysaccharide carbons. In addition, there are intramolecular cross peaks to aromatic lignin carbons in the range 115–153 ppm. The lack of substantial cross-peaks to Ac$^{Me}$, or to a polysaccharide carbon 1 suggests that significant cross-peaks between lignin CH$_3$O- and carbons with chemical shifts 62–90 ppm are only intramolecular at the 400 ms mixing time. However, at 1000 ms a cross-peak between the lignin CH$_3$O– and the GGM Ac$^{Me}$ appears. At 1500 ms, there are cross-peaks between the lignin CH$_3$O– and C1/Xn1$^{2f}$, M1$^{[Ac]}$, indicating that lignin and both acetylated and unacetylated mannosyl residues in GGM are spatially close. Nevertheless, at all mixing times, the majority of the cross-peaks from lignin CH$_3$O– are intramolecular, suggesting that most of the lignin is relatively distant from the polysaccharides.

To determine which of the various polysaccharides is closer to lignin we took slices at the carbon 4 shifts for GGM, xylan and cellulose domains 1C, 2B and 2C at the 1500 ms mixing time (Fig. 5c). The slices are normalised so that the carbon 4 self-peaks are all the same height. All five carbon 4 nuclei have a cross-peak to the lignin CH$_3$O– at 56.5 ppm, but there are significant differences between them. Relatively the strongest cross-peak detected is to M4$^{[Ac]}$, then Xn4$^{2f}$. The cross-peaks from C4$^{2B}$ and C4$^{2C}$ are of a similar intensity and the cross-peak to C4$^{1C}$ is almost undetectable. These results suggest lignin is most tightly associated with the xylan and GGM bound to the cellulose surface. There is also some lignin close to the cellulose surface, and this association is almost entirely with cellulose domain 2.

## Discussion

The molecular arrangement and interactions of lignin and polysaccharides in plant secondary cell walls is generally poorly understood due to technical difficulties of analysing this complex

material. Multidimensional solid-state NMR has recently been used to study the interactions of polymers in cell walls[12–14,23,37,40]. Here, we identified two-fold screw xylan in spruce, revealing that this mode of xylan interaction with the cellulose microfibril surface is a phylogenetically widespread feature of vascular plants. Furthermore, we also show that GGM is bound to the cellulose microfibril surface, and, we provide evidence to suggest that both hemicelluloses coat the surface of the same cellulose microfibrils. We also show that some lignin is closely associated with the cell wall polysaccharides, particularly xylan and GGM. These findings revise secondary cell wall models, contradict previous models of softwood molecular architecture, and will inform future studies that investigate how composite properties arise from nanoscale interactions.

Here we show that xylan exists in the two-fold screw conformation in conifer wood. This change in xylan conformation from the solution state three-fold screw indicates the conifer xylan is bound to the cellulose surface. The presence of strong xylan–cellulose cross-peaks in the CP-PDSD solid-state NMR spectra further shows that xylan does interact with the cellulose surface. The finding that gymnosperm xylan binds to cellulose contrasts with studies on conifers and the related gymnosperm, Ginkgo which suggested that xylan is not in contact with cellulose[7–9]. In Ginkgo, the xylan was proposed to bind cellulose-bound mannan perpendicular to the microfibril axis, but not directly on the microfibril surface. It is not possible to determine accurately by NMR the proportion of xylan that is bound to cellulose due to mobility differences that might affect detection of these xylan forms. However, comparing the intensity of the three-fold xylan and two-fold xylan peaks in CP-INADEQUATE experiments would suggest that the majority of the xylan is in the two-fold screw conformation, and thus the majority of xylan in spruce wood is bound to cellulose fibrils.

It has been previously shown that large portions of gymnosperm xylans have evenly spaced substitutions[22], which allow xylan to fold as a two-fold screw on the hydrophilic face of cellulose[14,23]. It seems likely therefore that much of the conifer xylan is bound to these faces of the fibril. However, in many plants including spruce there are significant regions of xylan with odd-spacing of substitutions[24,41]. In Arabidopsis, these are the glucuronic acid residues spaced 5, 6 and 7 xylose residues apart, while in spruce there are glucuronic acid residues on consecutive xylose residues. The presence of these patterns of xylan substitution, which are incompatible with binding to hydrophilic faces of cellulose as a two-fold screw, could lead to a proportion of xylan binding differently to cellulose or remaining unbound. The presence of three-fold xylan peaks in the solid-state CP NMR spectra support this suggestion of a matrix portion of xylan.

The molecular basis for interaction of GGM with cellulose has yet to be conclusively determined. Polarised FT-IR has previously suggested that GGM is aligned with the orientation of cellulose microfibrils in vivo[7,28]. One-dimensional solid-state MAS NMR experiments of Sitka spruce wood suggested that a fraction of hemicellulose is associated with cellulose, but the authors were unable to assign the hemicellulose signal to glucomannan and xylan[38]. To resolve these issues, we identified, using Arabidopsis molecular genetics and de novo assignments, the GGM backbone signals in spruce. We showed that GGM has strong cross-peaks with both cellulose domains in CP-PDSD experiments, suggesting the GGM is bound to the cellulose surface. As these cross-peaks appear at the same mixing time as the xylan–cellulose cross-peaks, both hemicelluloses are a similar distance from the microfibril.

Interestingly, the softwood GGM signals have a different $^{13}$C C4 chemical shift of 80.4 ppm when in the cell wall compared to that seen in solution. This is 2–13 ppm higher than the reported

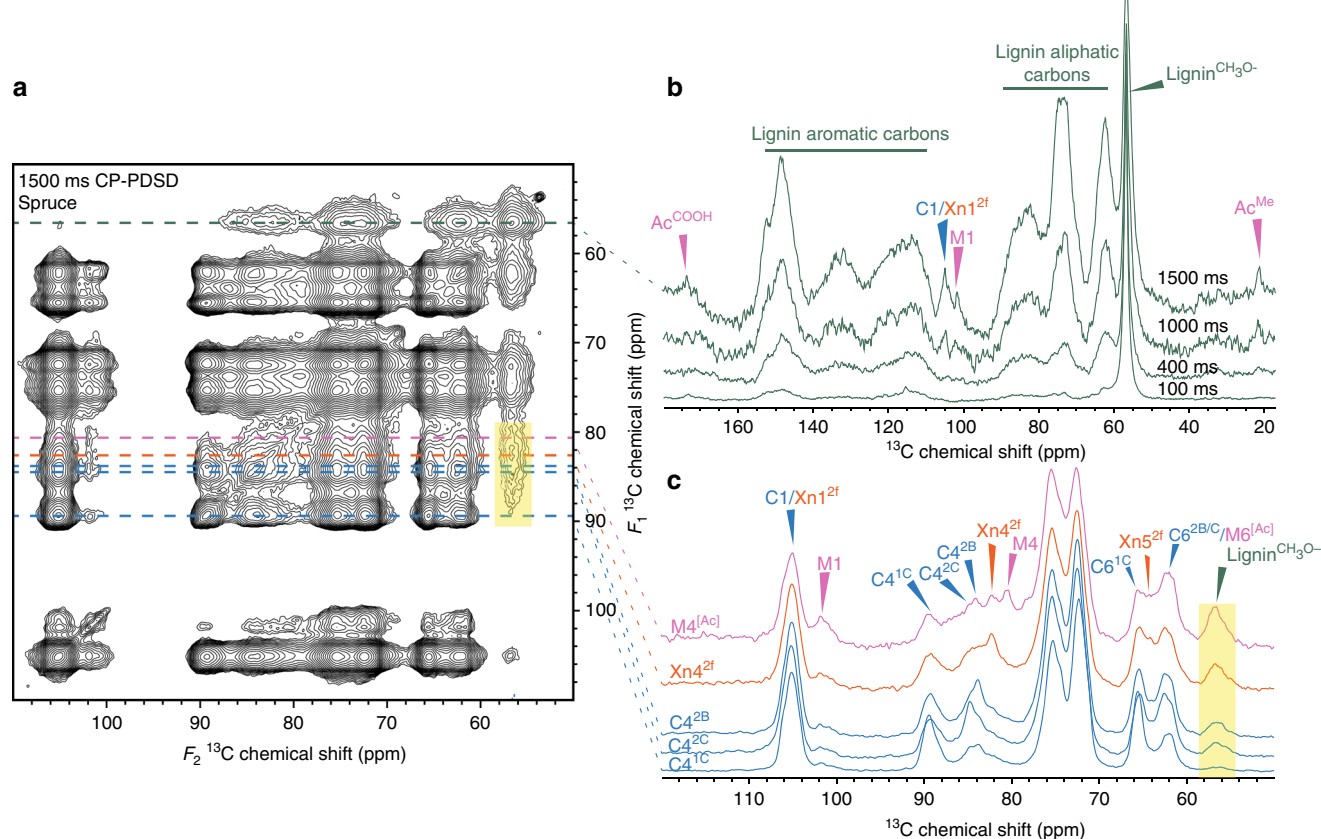

**Fig. 5** Lignin is more intimately associated with GGM and xylan than cellulose in spruce. **a** The carbohydrate region of a ${}^{13}$C MAS NMR 1500 ms mixing time CP-PDSD spectrum is shown, with lines drawn to mark where the 1D slices are derived from the 2D spectrum. **b** The lignin methoxyl (56.5 ppm) slice extracted from 2D CP-PDSD spectra at 100, 400, 1000 and 1500 ms mixing times are overlaid. The different spectra are normalised to the self-peak at 56.5 ppm. Cross-peaks between the lignin methoxyl and GGM acetate methyl and cellulose/xylan and GGM carbon 1 are labelled. **c** The carbon 4 slices of GGM (80.4 ppm), xylan (82.4 ppm) and three cellulose (2B = 83.7, 2C = 84.5, 1C = 89.5 ppm) sub-domains from the 1500 ms CP-PDSD spectrum are shown. Cross-peaks between the polysaccharides and the lignin methoxyl are labelled. All spectra are normalised so that the self-peak of carbon 4 is the same height. A translucent yellow box highlights the cross-peak to the lignin methoxyl

solution NMR chemical shifts for carbon 4 of mannosyl residues in GGM[26,28,35,42] (Supplementary Table 2). The M1 chemical shifts are slightly higher than in solution. The increased M4 shifts could suggest that GGM in muro has an altered C6 hydroxymethyl conformation compared to in solution[20,43]. However, C6 hydroxymethyl conformation has been shown in silico to alter significantly the chemical shifts of C2, C3, C5 and C6 in cellulose, as well as the shift of C4[19,20]. As our in muro M6[Ac] assignments are similar to those derived from solution NMR studies, hydroxymethyl conformation effects are an unlikely explanation for the observed GGM chemical shift differences. Previously, the glycosidic bond torsion angle has also been shown to have significant effects on carbon 1 and 4 chemical shifts in 1,4 linked sugars[14,44], and could be an explanation for the in muro GGM chemical shift changes. When unbranched mannans are crystallised in the mannan I allomorph, where they exist as a two-fold screw, the M1, M4 and M6 shifts are similar to our in muro GGM shifts[31,32]. Furthermore previous in vitro glucomannan/cellulose reconstitution studies have shown that glucomannan has a cellulose dependent change in M4 chemical shift[28]. It was proposed that GGM forms a two-fold screw on cellulose in vitro, due to the similarity of the GGM-cellulose composite chemical shifts to the mannan I crystal allomorph[28]. Combining these in vitro studies with our in muro results, we suggest that the GGM in spruce cell

walls adopts a two-fold screw-like conformation upon binding to the cellulose microfibril, but the precise interaction remains to be determined.

Since we have shown that both xylan and GGM bind to the surface of cellulose microfibrils, it is of considerable interest whether they bind to the same cellulose microfibrils, especially as they have previously been suggested to be spatially segregated[8,9]. We showed that there are cross-peaks between xylan and GGM in CP-PDSD experiments, most obviously at longer mixing times. This suggests that xylan and GGM are, at most, within about 5–10 Å of each other, and therefore mostly bound to the same or immediately adjacent cellulose microfibrils.

Solid-state NMR revealed that cellulose in spruce has multiple glucosyl residue environments, and that these can be placed into two main groups, which we call cellulose domains 1 and 2. Cellulose domain 1 has been proposed to comprise interior or crystalline cellulose, while domain 2 comprises surface glucan chains, based on numerous avenues of evidence[15,45]. The altered chemical shifts between domains 1 and 2 may arise from different hydroxymethyl conformation of the glucose units[15,20,21]. Modelling suggests that domain 1, in the interior of the microfibril, has the *tg* hydroxymethyl conformation due to hydrogen bonding by surrounding glucan chains, while the domain 2 glucosyl residues exist in the *gt* or *gg* conformation mostly on the

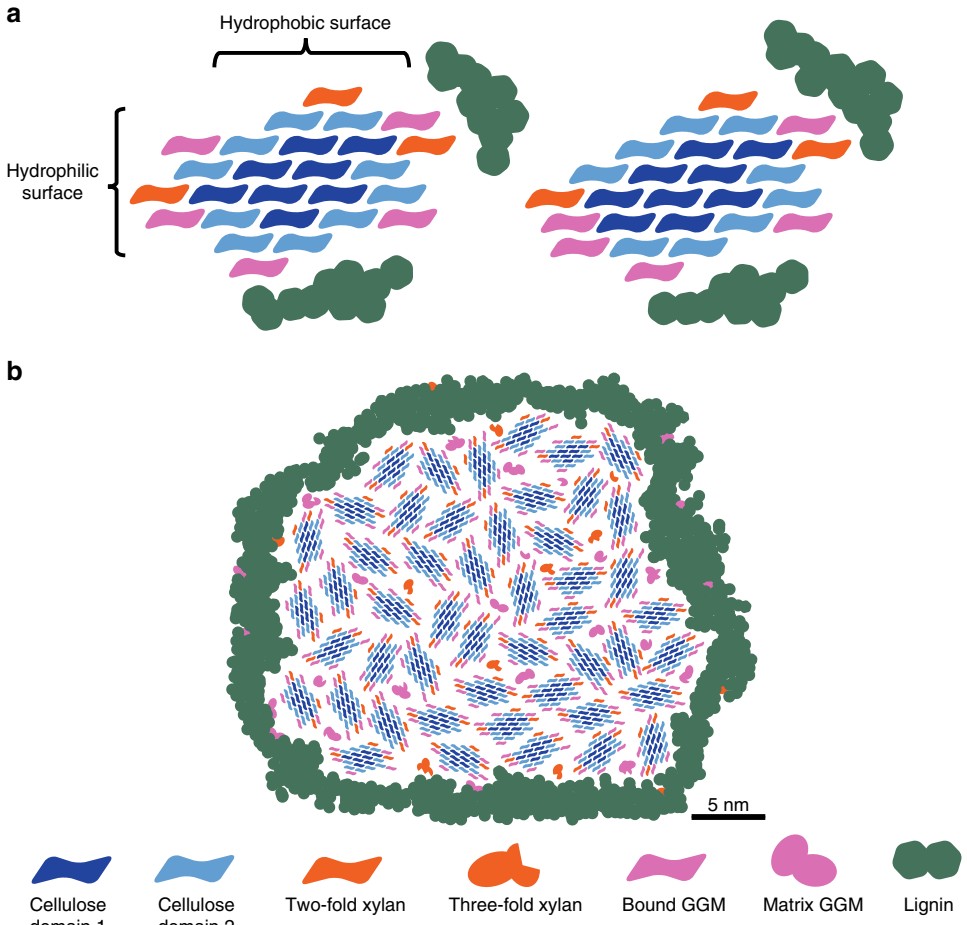

**Fig. 6 a** Possible models of the molecular architecture of softwood. The ratio of polysaccharide chains is based on the integrals of carbon 4–6 (carbon 4–5 for xylan) cross-peaks in the 30 ms CP-PDSD MAS NMR spectrum (see Fig. 2b) and the monosaccharide analysis. Both acetylated and unacetylated GGM are quantified together, as are the sub-domains of cellulose 1A–C and 2A–C. On the left, only xylan is shown as being able to convert domain 2 (C4 ≈ 84 ppm) to domain 1 (C4 ≈ 89 ppm) upon binding to the hydrophilic surface of the microfibril. On the right, binding of both xylan and some GGM to the hydrophilic surface can change domain 2 to domain 1. Lignin is shown mostly associated with itself, but is close to GGM, xylan and domain 2 cellulose. The cellulose microfibrils are taken to have 18 glucan chains with a 2, 3, 4, 4, 3, 2 habit to match the measured cellulose domain 1 to domain 2 ratio upon xylan binding. **b** Model of spruce cell wall macrofibril. Groups of cellulose microfibrils with bound GGM and xylan form macrofibrils in spruce cell walls. In addition to cellulose-bound xylan and GGM macrofibrils may contain some three-fold xylan and matrix GGM. Lignin is localised to the surface of the polysaccharide core of the macrofibril and interacts predominantly with GGM, xylan and cellulose domain 2. Size bar (5 nm) is provided for Fig. 6b) and is based on measurements presented in the literature[11, 16]

microfibril surface[21]. However, here we show that cellulose environment domains 1 and 2 are similarly close to GGM and xylan in spruce. Close proximity of xylan to cellulose domains 1 and 2 was also observed in Arabidopsis[14]. We propose that xylan binding to surface domain 2 chains, by hydrogen bonding in a manner that extends the crystalline arrangement of residues, might alter glucosyl hydroxymethyl conformation and hence convert the residues to cellulose domain 1[15] (Supplementary Fig. 6). In this model, domain 1 cellulose only occurs on the cellulose hydrophilic surfaces when bound to xylan. It seems possible that hydrophobic surface glucosyl residues maintain the *gg/gt* conformation (domain 2), but are bound by hemicelluloses through interactions mostly other than hydrogen bonding (Supplementary Fig. 6). Another potential explanation for the cross-peaks between hemicelluloses and domain 1 cellulose is that the hemicelluloses intercalate the cellulose microfibril, and thus have proximity to the interior domain 1 cellulose. However, if a complex substituted polymer could intercalate a crystalline cellulose microfibril, it would alter the bonding of adjacent glucan

chains and so probably change the NMR chemical shifts. Each of cellulose domains 1 and 2 comprises at least three glucosyl residue environments (1A-C, 2A-C, Supplementary Table 2). These may correspond to residues in different parts of the fibril or bound by hemicelluloses[46].

Previously, it has been suggested that lignin and xylan form a complex that is spatially separate from cellulose in conifer cell walls[7]. Other publications have suggested lignin is aligned parallel to the cellulose microfibril axis[8]. In silico studies have suggested that monolignols and dimers can bind to the hydrophobic face of cellulose microfibrils[47–49]. Here, we show clearly that lignin is associated with both xylan, GGM and domain 2 cellulose. This is similar to findings in a recent solid-state MAS NMR investigation of grass cell walls, where extensive non-covalent interactions between xylan and lignin were shown[37]. The proximity between lignin and hemicelluloses polysaccharide may be mediated by covalent linkages between the two, with a most notable softwood example being described very recently in pine where mannose M6 of GGM was demonstrated to be linked through an α-ether linkage to lignin[35].

Although we did not observe such covalent linkages in our data, the sensitivity of these solid-state NMR experiments is not yet high enough to identify these relatively rare structures.

Figure 6a incorporates our conclusions into a consistent model of softwood molecular architecture, which differs significantly from previous reports[7–9]. Our quantification of C4 peaks in 1D[13] and C4-C6 cross-peaks in a 2D 30 ms CP-PDSD spectrum indicates that, in spruce wood, domain 1 makes up ~42% and domain 2 constitutes 58% of the glucosyl residues in the microfibril. This corresponds to 8 domain 1 chains and 10 domain 2 chains in an 18-chain microfibril. We evaluated three potential arrangements of the 18 glucan chains (habits) of the cellulose microfibril which might be consistent with these proportions[46]. In the absence of any hemicelluloses, the most consistent cellulose microfibril habit is 34443[46]. However, binding of two-fold screw xylan onto the hydrophilic surface of the cellulose microfibril[14,22,23,25] induces transition of domain 2 glucosyl residues to domain 1. This would lead to formation of more domain 1 in the 34443 microfibril than consistent with the NMR measurements. In light of this result, we re-evaluated additional 18-chain microfibril habits[46]. Kubicki et al., suggested that the 234432 model may exist in muro but it was disfavoured due to the presence of only 6 domain 1 glucans[13,46]. However, our results suggest a hemicellulose driven domain 2 to domain 1 conversion so the 234432 model with two xylan chains bound on the hydrophilic surface satisfies better the ratio of the two cellulose environments. Therefore, in our model, we show domain 1 and domain 2 glucan chains at both the interior and surface of the 234432 microfibril, with domain 1 cellulose only occurring on the surface when bound by a xylan chain (left hand side Fig. 6a). It is possible some GGM may also alter the cellulose conformation from domain 2 to domain 1. This accounts for the similar proximity of GGM to both domain 1 and domain 2 but would lead to some variation from the observed cellulose domain ratio (right hand side Fig. 6a). The basis of this discrepancy is unclear, but might be explained by imperfections of the microfibril crystal leading to some glucan or hemicellulose chains dissociating from the hydrogen bonding network. In both models, lignin is shown nearby to hemicelluloses and to some domain 2 glucan chains, consistent with the PDSD experiments.

The hemicellulose-coated elementary cellulose fibrils and lignin assemble into larger order structures in softwood[10,11]. These macrofibrils have a median diameter of approximately 30 nm and therefore contain tens of individual cellulose microfibrils. The distance between individual microfibrils is likely to accommodate only one hemicellulose chain[15,16]. Our data are consistent with this model, where lignin is mainly localised on the surface of a polysaccharide macrofibril core and some GGM and three-fold xylan exists as matrix hemicellulose not interacting directly with the microfibril (Fig. 6b). The data presented here provides constraints on models of softwood cell wall architecture by direct measurement of molecular proximities using solid-state NMR. In the future such constraints could be used to inform molecular dynamics simulations of polysaccharide interactions in spruce cell walls, to uncover more details of cell wall interactions[30]. Thus, our work contributes significantly to the understanding of cell wall structure and can influence approaches to the traditional uses of softwood, such as in the construction industry, or in novel processes, such as in nanocellulose manufacturing.

## Materials and methods

**Plant material**. [13]C enriched, never-dried, de-barked *Picea abies* (Norway spruce) softwood was purchased from IsoLife BV (Wageningen Plant Research, Wageningen UR, The Netherlands). Spruce wood sections (20 mm length, 3–5 mm diameter) were frozen in liquid nitrogen and ground using ceramic mortar and pestle to produce a fine powder. WT and *csla2csla3csla9 Arabidopsis thaliana* material was grown in an in-house built [13]C enrichment chamber[14,39]. In all, 30–50 mg of the plant material was packed into a 3.2 mm MAS rotor for NMR experiments.

**Determination of [13]C enrichment**. Alcohol insoluble residue (AIR) was prepared sections of *Picea glauca* and [13]C enriched *P. abies* wood. All AIR preparation was carried out as described in[50]. Aliquots (0.5 mg) of *P. glauca* and *P. abies* AIR were pre-treated with 4 M NaOH, neutralised, re-suspended in 0.1 M Ammonium Acetate buffer pH = 5.5 and digested with *Neocallimastix patriciarum* xylanase GH11.

Released oligosaccharides were dried and re-suspended in 20 μl water. 1 μl of the oligosaccharide solution was mixed with an equal volume of 20 mg ml$^{-1}$ 2,5-dihydroxybenzoic acid (DHB) in 50% acetonitrile, 0.1% TFA on a SCOUT-MTP 384 target plate (Bruker)[51]. The spotted samples were dried in a vacuum dessicator. The samples were analyzed by mass spectrometry on an Ultraflex III matrix-assisted laser desorption ionisation-time of flight/time of flight (MALDI/TOF-TOF) instrument (Bruker). The data were collected with a 2-kHz smartbeam-II laser and acquired on reflector mode, the mass range was 300–3000 Da. Data acquisition and analysis was performed in FlexControl and FlexAnalysis software respectively. On average, about 10,000 shots were used to obtain the spectra[52]. The degree of [13]C enrichment in *P. abies* softwood was quantified by integration of MS peak volume of fully and partially enriched oligosaccharides[14].

Based on the masses and peak heights in MALDI-TOF spectra of GH11 xylanase generated xylo-oligosaccharides, which can be used to calculate the number of [12]C and [13]C atoms in the oligosaccharide, we calculated the [13]C enrichment of xylan in this sample to be over 97%. See Supplementary Fig. 3.

**Monosaccharide analysis**. [13]C enriched *P. abies* wood AIR was incubated with 2 M TFA for 1 h at 120 °C to hydrolyse the polysaccharides into monosaccharides. Following evaporation under vacuum for the removal of TFA, the sample was resuspended in 200 μl of water and the monosaccharide sugars were separated using protocols adapted from Currie and Perry on a Dionex ICS3000 system equipped with a Carbopac PA20 column[53].

**Solid-state NMR of [13]C enriched *P. abies* and *A. thaliana***. Solid-state MAS NMR experiments were performed on Bruker (Karlsruhe, Germany) Advance III solid-state NMR spectrometers, operating at [1]H and [13]C Larmor frequencies of 850.2 and 213.8 MHz and 700 and 176.0 MHz, respectively, using 3.2 mm double-resonance MAS probes. Experiments were conducted at room temperature at MAS frequencies of 12.5 kHz on the 850 MHz spectrometer and 12 kHz on the 700 MHz spectrometer. The [13]C chemical shift was determined using the carbonyl peak at 177.8 p.p.m. of L-alanine as an external reference with respect to tetramethylsilane (TMS); 90° pulse lengths were 2.65–3.5 μs ([1]H) and 4.0 μs ([13]C). Both [1]H–[13]C cross-polarisation (CP) with ramped (70–100%)[1]H rf amplitude and, typically, 1 ms contact time and direct polarisation (DP) were used to obtain the initial transverse magnetisation[54]. While CP emphasises the more rigid material, a short, 1.9 s, recycle delay DP experiment was used to preferentially detect the mobile components and a 20 s delay was used for quantitative experiments. SPINAL-64 decoupling[55] was applied during acquisition at a [1]H nutation frequency of 70–80 kHz. The [13]C spin lattice relaxation time, $T_1$, was measured at 176.0 MHz using saturation recovery following a comb of 30 pulses and echo acquisition. The [1]H homonuclear coupling was suppressed using the frequency switched Lee-Goldberg sequence with a [1]H nutation frequency of 80 kHz[56]. Two-dimensional double-quantum (DQ) correlation spectra were recorded using the refocused INADEQUATE pulse sequence which relies upon the use of isotropic, scalar J coupling to obtain through-bond information regarding directly coupled nuclei[57–59]. The carbon 90° and 180° pulse lengths were 4 and 8 μs, respectively with 2τ spin-echo evolution times for a (π–τ–π/2) spin-echo of 4.48 ms and the SPINAL-64 [1]H decoupling was applied during both the evolution and signal acquisition periods[55]. Intermolecular contacts were probed using 2D [13]C–[13]C proton driven spin diffusion (PDSD) experiments with mixing times of 30 ms to 1.5 s[60,61]. The acquisition time in the indirect dimension ($t_1$) was 5.0 ms in the CP-INADEQUATE and 7–10 ms in the CP-PDSD experiments. The sweep width in the indirect dimension was 50 kHz for both experiments with 128 acquisitions per $t_1$ for the CP-INADEQUATE and 16-48 acquisitions for the CP PDSD experiments. The recycle delay was 2 s. For both refocused INADEQUATE and PDSD experiments, the spectra were obtained by Fourier transformation into 4 K ($F_2$) × 2 K ($F_1$) points with exponential line broadening of 40 Hz (CP) in $F_2$ and squared sine bell processing in $F_1$. All spectra obtained were processed and analysed using Bruker Topspin version 3.5 pl 7.

## Data availability

Unprocessed NMR data files available from https://doi.org/10.17863/CAM.43404.

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

## Acknowledgements

We would like to thank Michael Cafferkey for his contribution to Fig. 6 and to Supplementary Fig. 6, IsoLife for $^{13}C$ enriched spruce and Mike Jarvis for useful discussions on cellulose structure. O.M.T was a recipient of an iCASE studentship from the BBSRC and Novozymes (Reference BB/M015432/1). J.J.L. was in receipt of a studentship from the BBSRC as part of the Cambridge BBSRC-DTP Programme (Reference BB/J014540/1). L.Y. and P.D. are supported by the Leverhulme Trust Natural Material Innovation Centre. The UK 850 MHz solid-state NMR Facility used in this research was funded by EPSRC and BBSRC (contract reference PR140003), as well as the University of Warwick including via part funding through Birmingham Science City Advanced Materials Projects 1 and 2 supported by Advantage West Midlands (AWM) and the European Regional Development Fund (ERDF).

## Author contributions

O.M.T, J.J.L, L.Y, D.I, T.W.F and R.D designed and performed experiments. O.M.T, J.J.L, S.P.B, R.D and P.D analysed data and wrote the paper.

## Competing interests

The authors declare no competing interests.
