## [Peer Review File · Nature Communications]

Reviewers' comments:

Reviewer #1 (Remarks to the Author):

This is a beautiful work on a very relevant and still obscure issue related to understanding of the cell wall assembly and subnanometer interactions between cell wall polymers. The present study chose an approach (solid state NMR) that allows study of the never dried wood and avoids any possible artifact arising from isolation and sampling issues. The conclusions are fully supported by clear and straightforward experimental evidence. The assignment of cellulose, xylan and galactoglucomannan in solid state NMR is convincing. The proximity interactions between different biopolymers are clearly and unambiguously defined. In this context the role of lignin-polysaccharides interactions is extremely relevant. Intermolecular proximity between lignin and xylan/galactoglucomannan was clearly defined in the present study by identification of cross peaks at 1500 and 1000 ms mixing times. However the authors are not clear enough when referring to cross peaks with lower mixing times. In page 12 they write: "...Until 1000 ms the only cross-peaks are probably intermolecular, from the lignin CH₃O- to aliphatic lignin carbons in the range 62-90 ppm that are chemically similar to polysaccharide carbons. In addition there are intramolecular cross peaks to aromatic lignin carbons Nevertheless, at all mixing times, the majority of cross peaks from lignin CH₃O- are intramolecular, suggesting that most of the lignin is relatively distant from the polysaccharides". It is not clear whether there are cross peaks between the lignin CH₃O- and polysaccharides signals at low mixing times or not. The authors should clearly discuss this point since it is related to the important issue of the presence and nature of LCCs in wood with extremely relevant implications. Also in page 18 the authors write that "the proximity between lignin and hemicelluloses polysaccharide may be mediated by covalent linkages between the two". Here they should also clearly state whether they have experimental evidence of that or not.

Reviewer #2 (Remarks to the Author):

Nature Communications manuscript : # NCOMMS-19-17369
Molecular architecture of softwood revealed by solid-state NMR
Oliver M. Terrett, Jan J. Lczakowski, Li Yu, Dinu Luga, Trent W. Franks, Steven P. Brown, Ray Dupree and Paul Dupree

In this work the authors, using ¹³C multidimensional solid-state NMR, study the interactions between the different biopolymers (celluloses, lignins and hemicelluloses) of the cell wall of softwood. Chemical changes, through bonds and through space interactions were mainly investigated. Then, they propose a revised model of cell wall which matches the NMR results.

This manuscript, based on rather classical NMR strategy, is clear and well presented. The different experiments are well described the results and conclusions are clear. As a matter of fact it is fully relevant for publication in Nature Communications.

I just want to mention some minor points or questions which may be relevant:

1. The authors use several times «in muro» expression within the text. It may be interesting to precise (or remind) the exact meaning they want to introduce since I am not sure that it is perfectly understandable for everyone?
2. I don't understand the sentence "the enrichment was calculated to be more than 97% ¹³C, see supplementary figure 3" I suppose it is related to ref 14. Make it clearer.
3. It can be good to use "13C enriched" instead of "13C labelled" to avoid confusion with labelling (numbering) of carbons or residues
4. Page 1, « This is 2-13 ppm higher than the reported solution NMR... » how do the authors explain the 13 ppm limit which seems rather difficult to justify on the basis of crystallographic and/or conformational changes. Does it mean that "in muro" interactions with celluloses such as hydrogen

bonding may be also involved and explain such chemical shifts changes.

5. Page 16, it seems than a word is missing in the sentence « The M1 chemical shifts are slightly than ??? in solution ».

6. The authors assign the different chemical shifts of 4C of cellulose microfibers to inside and outside glucosyl residues. Is it the only possibility?

7. Most of the results concerning the spatial distribution of the different wood components are based on the occurrence or the absence of dipolar ^{13}C - ^{13}C interactions between the different cell wall components. The specific molecular dynamics of each of them can be very different ranging from rigid polymers (e.g. crystalline celluloses) to more flexible polymers which may modify the intensity of the correlations. Is it possible to take into account these features for quantitative interpretation?

8. Base on these results is there a possibility to use such data to run computational simulations in order to propose a refined 3D model of the cell wall? A comment may be added in the discussion or conclusion sections.

Reviewer #1 (Remarks to the Author):

This is a beautiful work on a very relevant and still obscure issue related to understanding of the cell wall assembly and subnanometer interactions between cell wall polymers. The present study chose an approach (solid state NMR) that allows study of the never dried wood and avoids any possible artifact arising from isolation and sampling issues. The conclusions are fully supported by clear and straightforward experimental evidence. The assignment of cellulose, xylan and galactoglucomannan in solid state NMR is convincing. The proximity interactions between different biopolymers are clearly and unambiguously defined.

Response: Thank you for your positive and constructive comments about the paper. We have added sentences to make our meaning clearer where you have requested this as described below.

In this context the role of lignin-polysaccharides interactions is extremely relevant. Intermolecular proximity between lignin and xylan/galactoglucomannan was clearly defined in the present study by identification of cross peaks at 1500 and 1000 ms mixing times. However the authors are not clear enough when referring to cross peaks with lower mixing times. In page 12 they write: "...Until 1000 ms the only cross-peaks are probably intermolecular, from the lignin CH₃O- to aliphatic lignin carbons in the range 62-90 ppm that are chemically similar to polysaccharide carbons. In addition there are intramolecular cross peaks to aromatic lignin carbons ...Nevertheless, at all mixing times, the majority of cross peaks from lignin CH₃O- are intramolecular, suggesting that most of the lignin is relatively distant from the polysaccharides". It is not clear whether there are cross peaks between the lignin CH₃O- and polysaccharides signals at low mixing times or not.

Response: We have added the following sentence on line 266 "The lack of substantial cross-peaks to Ac^{Me}, or to a polysaccharide carbon 1 suggests that significant cross-peaks between lignin CH₃O- and carbons with chemical shifts 62-90 ppm are only intramolecular at the 400 ms mixing time."

We hope this makes it clearer that at the shorter mixing times of 100 ms and 400 ms the cross-peaks are likely to be only intramolecular within lignin. At 1000 ms and 1500 ms unambiguous cross-peaks to polysaccharide carbons (Ac^{Me} at 1000 ms and C1/Xn1^{2f} at 1500ms) appear suggesting intermolecular proximity between some lignin and the polysaccharides.

The authors should clearly discuss this point since it is related to the important issue of the presence and nature of LCCs in wood with extremely relevant implications. Also in page 18 the authors write that "the proximity between lignin and hemicelluloses polysaccharide may be mediated by covalent linkages between the two". Here they should also clearly state whether they have experimental evidence of that or not.

Response: Unfortunately, the sensitivity of these experiments is not yet high enough to demonstrate the presence of such linkages in native wood, which based on Nishimura et al., (2018) may be very infrequent. We have added a sentence on line 397 to make this clearer ("Although we did not observe such covalent linkages in our data, the sensitivity of these solid-state NMR experiments is not yet high enough to identify these relatively rare structures."). However, such linkages may be very important for maintaining the proximity between lignin and polysaccharides in the cell wall.

Reviewer #2 (Remarks to the Author):

Nature Communications manuscript : # NCOMMS-19-17369

Molecular architecture of softwood revealed by solid-state NMR

Oliver M. Terrett, Jan J. Lczakowski, Li Yu, Dinu Luga, Trent W. Franks, Steven P. Brown, Ray Dupree and Paul Dupree

In this work the authors, using ^{13}C multidimensional solid-state NMR, study the interactions between the different biopolymers (celluloses, lignins and hemicelluloses) of the cell wall of softwood. Chemical changes, through bonds and through space interactions were mainly investigated. Then, they propose a revised model of well wall which matches the NMR results. This manuscript, based on rather classical NMR strategy, is clear and well presented. The different experiments are well described the results and conclusions are clear. As a matter of fact it is fully relevant for publication in Nature Communications.

Response: Thank you for your positive and constructive comments about the paper. We have answered your questions below, and have adapted certain sections to reflect your suggested changes.

I just want to mention some minor points or questions which may be relevant:

1. The authors use several times «in muro» expression within the text. It may be interesting to precise (or remind) the exact meaning they want to introduce since I am not sure that it is perfectly understandable for everyone?

Response: This term arises from Latin for wall (murus). On the first mentioning of the phrase *in muro*, line 110, we have added the following sentence. “Here, *in muro*, refers to the native location of wood polysaccharides and lignin in the plant cell wall, in contrast to extracted *in vitro* polysaccharides.”

2. I don’t understand the sentence “the enrichment was calculated to be more than 97% ^{13}C , see supplementary figure 3” I suppose it is related to ref 14. Make it clearer.

Response: We have altered this sentence to the following, to make the meaning clearer, “Based on the masses, which enable calculation of the number of ^{12}C versus ^{13}C atoms, and peak area in MALDI-TOF spectra of GH11 xylanase generated xylo-oligosaccharides, we calculated the ^{13}C enrichment of xylan in this sample to be over 97%.” We assume the ^{13}C enrichment of all polysaccharides and lignin is equivalent to the ^{13}C enrichment of xylan.

3. It can be good to used “ ^{13}C enriched” instead of “ ^{13}C labelled” to avoid confusion with labelling (numbering) of carbons or residues

Response: We have changed all instances of “labelled” to “enriched”.

4. Page 1, « This is 2-13 ppm higher than the reported solution NMR... » how do the authors explain the 13 ppm limit which seen rather difficult to justify on on the base of crystallographic and/or

conformational changes. Does it mean that “in muro” interactions with celluloses such as hydrogen bonding may be also involved and explain such chemical shifts changes.

Response: We agree the crystallographic and conformational changes are unlikely to explain such a large difference, and we suspect that the previous literature may contain a mis-assigned mannosyl residue M4 shift. The reference (Hannuksela et al., 2004) assigns just one minor type of M4 as 67 ppm. This minor mannosyl residue assigned as an unsubstituted mannosyl residue adjacent to a 2-acetylated mannosyl residue and a glucosyl residue.

All other mannosyl residues in solution have M4 shifts between 74.2–78.5 ppm (Hannuksela et al., 2004), and (Nishimura et al 2018) report 77.6-78.0 ppm as the range for their M4 shifts. Thus most mannosyl residues in solution NMR studies have a M4 chemical shift that is only 2-6 ppm lower than our *in muro* M4 chemical shift. We would agree with the reviewer that the increase from 74.2-78.5 ppm to our reported *in muro* M4 chemical shift of 80.4 ppm may be caused by conformational changes. Supporting this, the two cellulose domains, which have C4 chemical shift differences of 4 ppm (similar to the difference of the solution M4 values versus our reported M4 values), are thought to differ in the conformation of the C6 hydroxymethyl group (Jarvis, 2018). There could also be some effect on the M4 chemical shift from hydrogen bonding from cellulose to GGM carbons, and this hydrogen bonding may also facilitate the conformational changes which we have suggested occur.

5. Page 16, it seems than a word is missing in the sentence « The M1 chemical shifts are slightly than ??? in solution ».

Response: We have amended the sentence to say “slightly higher”.

6. The authors assign the different chemical shifts of 4C of cellulose microfibrils to inside and outside glucosyl residues. Is it the only possibility?

Response: This is a difficult and important issue that we discuss in the manuscript. On page 17 we describe previous conclusions in the field, where the domain 1 and domain 2 glucan chains of cellulose have been proposed to correspond to the interior and surface glucan chains in a cellulose microfibril. The C4 chemical shift differences have been suggested to be due to the different *tg* and *gg/gt* carbon 6 hydroxymethyl conformations.

Our results contrast with some of these previous results. Here we conclude that the domain 1 chains (previously described as interior) are also found on the microfibril surface. This is because the domain 1 glucan chains have cross-peaks to xylan and mannan, which can presumably only bind to the cellulose surface. We propose that binding of hemicelluloses might convert some domain 2 glucan chains on the microfibril surface to domain 1, by altering the carbon 6 hydroxymethyl conformation through hydrogen bonding. Thus it seems that glucan chains in domain 1 and 2 can occur on the surface of the cellulose microfibril.

To be consistent with our results, in our model in figure 6 we show some domain 1 and 2 chains on the surface and interior of the cellulose microfibril.

7. Most of the results concerning the spatial distribution of the different wood components are based on the occurrence or the absence of dipolar ^{13}C - ^{13}C interactions between the different cell wall components. The specific molecular dynamics of each of them can be very different ranging from rigid polymers (e.g. crystalline celluloses) to more flexible polymers which may modify the

intensity of the correlations. Is it possible to take into account these features for quantitative interpretation?

Response: We agree that polymers with different mobilities may produce different strengths of cross-peaks in our PDS experiments. For instance, very mobile hemicelluloses could be close to cellulose, but may not produce cross-peaks in CP-PDS experiments as CP excitation is inefficient for highly mobile polymers. For instance, the terminal arabinose peaks we identified in the CP-INADEQUATE seem to have very few intermolecular peaks in CP-PDS experiments, and this may be due to high mobility.

To control for differences in polysaccharide mobility we measured the T₁ relaxation times for different carbons of different polymers in spruce, and this is shown in supplementary table 3. Although some carbons (e.g Ac^{Me}) have multicomponent T₁ relaxation curves, for the majority of polysaccharide carbons the largest component of the T₁ relaxation times are long, at between 5-6 s. This is similar to the T₁ of cellulose carbons (between 5-6 s for this sample). Thus it is unlikely that the proximity of xylan and mannan to cellulose is “suppressed” by high mobility of these hemicelluloses.

The lignin has slightly shorter T₁ times at 3 s, and the size of intermolecular cross-peaks from lignin to polysaccharides may be slightly reduced due to this effect.

In our figure 6 we have shown the relative amounts of different polysaccharides based on their relative signal strengths in short mixing time PDS experiments and monosaccharide analysis. Multidimensional NMR experiments are not fully quantitative, so this is an estimate. It would increase the uncertainty of these estimates to try to control for slightly different T₁ relaxation times of different carbons. Figure 6 is only intended to be a model of the spruce cell wall on the nanoscale and should not be considered to reflect accurate quantities.

8. Based on these results is there a possibility to use such data to run computational simulations in order to propose a refined 3D model of the cell wall? A comment may be added in the discussion or conclusion sections.

Response: This is an excellent suggestion for future work. It would be very interesting to compare the inter-atomic distances between xylan, mannan, cellulose and lignin in computer simulations with the constraints of solid-state NMR to try and unravel more details of the interactions between different cell wall components. We have included the following sentence on line 434 to emphasise that further work is required, especially in relating solid-state NMR proximity data to detailed molecular models. “In the future such constraints could be used to inform molecular dynamics simulations of polysaccharide interactions in spruce cell walls, to uncover more details of cell wall interactions”.

In addition to the reviewers comments the following changes to manuscript have been made:

Introduction shortened to comply more closely with word limit.

Corresponding author email addresses added.

Results section sub-headings reduced to reflect character limits.

Data availability section added.

References edited to comply with journal guidelines.

Main Text Table altered to comply with guidelines.

References added to supplementary information.

REVIEWERS' COMMENTS:

Reviewer #1 (Remarks to the Author):

The modifications made by the authors to the first draft are appropriately addressing the reviewer's concerns. Therefore I recommend publication without any further modification

Reviewer #2 (Remarks to the Author):

I don't see any remaining problems or questions to be addressed.
All the remarks were fully answered and the manuscript was consequently amended. Thus I do recommend to accept it.
Michel Bardet